# BHLHE41/DEC2 Expression Induces Autophagic Cell Death in Lung Cancer Cells and Is Associated with Favorable Prognosis for Patients with Lung Adenocarcinoma

**DOI:** 10.3390/ijms222111509

**Published:** 2021-10-26

**Authors:** Toshiyuki Nagata, Kentaro Minami, Masatatsu Yamamoto, Tsubasa Hiraki, Masashi Idogawa, Katsumi Fujimoto, Shun Kageyama, Kazuhiro Tabata, Kohichi Kawahara, Kazuhiro Ueda, Ryuji Ikeda, Yukio Kato, Masaaki Komatsu, Akihide Tanimoto, Tatsuhiko Furukawa, Masami Sato

**Affiliations:** 1Department of General Thoracic Surgery, Graduate School Medical and Dental Sciences, Kagoshima University, 8-35-1 Sakuragaoka, Kagoshima 890-8544, Japan; t.nagata@scchr.jp (T.N.); k7433286@kadai.jp (K.U.); masa3310@m2.kufm.kagoshima-u.ac.jp (M.S.); 2Department of Molecular Oncology, Graduate School Medical and Dental Sciences, Kagoshima University, 8-35-1 Sakuragaoka, Kagoshima 890-8544, Japan; kentaro_minami@med.miyazaki-u.ac.jp (K.M.); masatatu@m2.kufm.kagoshima-u.ac.jp (M.Y.); k-kawahr@m3.kufm.kagoshima-u.ac.jp (K.K.); 3Department of Pharmacy, University of Miyazaki Hospital, 5200 Kihara Kiyotake cho, Miyazaki 889-1692, Japan; ryuji_ikeda@med.miyazaki-u.ac.jp; 4Department of Pathology, Graduate School Medical and Dental Sciences, Kagoshima University, 8-35-1 Sakuragaoka, Kagoshima 890-8544, Japan; tsubasa0121h@gmail.com (T.H.); tabatak.kufm@gmail.com (K.T.); akit09@m3.kufm.kagoshima-u.ac.jp (A.T.); 5Department of Medical Genome Sciences, Research Institute for Frontier Medicine, Sapporo Medical University, Sapporo 060-8556, Japan; idogawa@sapmed.ac.jp; 6Department of Dental and Medical Biochemistry, Institute of Biomedical and Health Sciences, Hiroshima University, Hiroshima 734-8553, Japan; kfujimo@hiroshima-u.ac.jp (K.F.); ykato@hiroshima-u.ac.jp (Y.K.); 7Department of Physiology, Juntendo University Graduate School of Medicine, Bunkyo-ku, Tokyo 113-8421, Japan; s.kageyama.gk@juntendo.ac.jp (S.K.); mkomatsu@juntendo.ac.jp (M.K.); 8Center for the Research of Advanced Diagnosis and Therapy of Cancer, Graduate School of Medical and Dental Sciences, Kagoshima University, 8-35-1 Sakuragaoka, Kagoshima 890-8544, Japan

**Keywords:** non-small cell lung cancer, BHLHE41, autophagic cell death, non-invasiveness

## Abstract

Lung cancer constitutes a threat to human health. BHLHE41 plays important roles in circadian rhythm and cell differentiation as a negative regulatory transcription factor. This study investigates the role of BHLHE41 in lung cancer progression. We analyzed BHLHE41 function via in silico and immunohistochemical studies of 177 surgically resected non-small cell lung cancer (NSCLC) samples and 18 early lung squamous cell carcinoma (LUSC) cases. We also examined doxycycline (DOX)-inducible BHLHE41-expressing A549 and H2030 adenocarcinoma cells. BHLHE41 expression was higher in normal lung than in lung adenocarcinoma (LUAD) tissues and was associated with better prognosis for the overall survival (OS) of patients. In total, 15 of 132 LUAD tissues expressed BHLHE41 in normal lung epithelial cells. Staining was mainly observed in adenocarcinoma in situ and the lepidic growth part of invasive cancer tissue. BHLHE41 expression constituted a favorable prognostic factor for OS (*p* = 0.049) and cause-specific survival (*p* = 0.042) in patients with LUAD. During early LUSC, 7 of 18 cases expressed BHLHE41, and this expression was inversely correlated with the depth of invasion. DOX suppressed cell proliferation and increased the autophagy protein LC3, while chloroquine enhanced LC3 accumulation and suppressed cell death. In a xenograft model, DOX suppressed tumor growth. Our results indicate that BHLHE41 expression prevents early lung tumor malignant progression by inducing autophagic cell death in NSCLC.

## 1. Introduction

Lung cancer is the most common cause of tumor-related death worldwide [1]. However, prognostic evaluation of non-small cell lung cancer (NSCLC) remains primarily based on clinical parameters [2,3], despite the rapid progression in our understanding of the genetic changes in lung cancers [4,5]. Therefore, it is crucial to explore the molecules that contribute to cancer progression and patient prognosis in order to understand the mechanisms of progression, predict tumor aggressiveness, and identify better guides for clinical treatment [6].

BHLHE41, also known as DEC2/SHARP-1/ BHLHB3, is a helix loop helix suppressive transcription factor that was identified based on its sequence similarity to BHLHE40/DEC1/SHARP-2/STRA-13 [7]. BHLHE41 functions as a negative regulator of circadian rhythm suppression of Clock/Bmal1, Per expression, similar to BHLHE40 [8] and regulates sleep duration. DEC2(P385R) genotype is associated with a human short sleep phenotype [9]. In cell differentiation, BHLHE41 can interact with C⁄EBPα and β, inhibit their transcriptional activity and suppress adipogenesis in adipocyte precursor cells [10]. BHLHE41 has been also reported to express selectively in Th2 cells among Th lineage and indispensable to Th2 differentiation in mice [11]. The regulation of expression of multiple genes by BHLHE40 and BHLHE41 also has been reported to affect cancer development in apoptosis, epithelial mesenchymal transition, and hypoxic reaction [12]. BHLHE41 expression suppresses apoptosis induced by TNFα in breast cancer cells [13], TWIST1 expression and migration in human endometrial cancer cells [14], is also associated with better prognosis of patients with breast cancer with HIF1 protein suppression [15]. It has also been reported to have oncogenic roles in renal clear-cell cancer and MLL-AF6-positive acute myelogenous leukemia (AML) with other transactivating proteins [16,17]. From these previous studies, BHLHE41 can work as both a tumor promoter and suppressor, depending on the cellular context. BHLHE41 also has been reported to act as a tumor suppressor gene in the lung [18], however, the roles of BHLHE41 in lung cancer development remain unclear. In the present study, we, therefore, examine the correlation between BHLHE41 expression and the prognosis of patients with NSCLC and explored the potential roles of BHLHE41 as a tumor suppressor during lung cancer development and progression.

## 2. Results

### 2.1. BHLHE41 Expression Is Associated with Favorable Prognosis for Patients with Lung Adenocarcinoma (LUAD) as Shown by In Silico and Immunohistochemistry (IHC) Analyses 

First, to determine whether BHLHE41 expression is related to the outcome of patients with lung cancer, we performed a Kaplan–Meier analysis of overall survival (OS) using The Cancer Genome Atlas (TCGA) data. BHLHE41 expression was significantly higher in normal lung than in LUAD and lung squamous cell carcinoma (LUSC) (*p* = 1.74 × 10^−11^ and *p* = 1.47 × 10^−7^, respectively) (Figure 1a,b). The high BHLHE41 expression group showed a significantly favorable outcome compared to that of the low expression group for LUAD (*p* = 0.00992) but not for LUSC (*p* = 0.07403) (Figure 1c and Appendix A).

To determine whether BHLHE41 protein expression is associated with the prognosis of patients with lung cancer, we performed IHC analysis of clinical specimens from 177 lung cancer cases (Table 1). Normal alveolar epithelial cells were clearly stained in the nuclei (Figure 2a). In LUAD, BHLHE41 mainly stained the nuclei of several tumors diagnosed as adenocarcinoma in situ (AIS) and the lepidic growth part of lepidic-predominant cancer (Figure 2b,c). Some lepidic growth parts and the most invasive portion of cancer cells were not stained (Figure 2d,e). Only 2 out of 45 (4.4%) non-LUAD tumors, including 43 squamous cell carcinomas and 2 pleomorphic carcinomas, showed positive staining (Figure 2f), and the remaining cancers did not express BHLHE41 (Figure 2g). 

A total of 15 out of 132 (11.4%) patients with LUAD and 17 of 177 (9.6%) with NSCLC were assessed as being BHLHE41 positive (Table 2 and Appendix A). The patients were then divided into two groups based on BHLHE41 staining and evaluated for correlation with prognosis. Patients in the BHLHE41-positive group with NSCLC and LUAD showed significantly more favorable OS (*p* = 0.040 and *p* = 0.049, respectively) and cause-specific survival (CS) (*p* = 0.024 and *p* = 0.042, respectively) than those with BHLHE41-negative cancer (Figure 3a–d). In comparison, in the non-LUAD group, it was difficult to evaluate association between BHLHE41 expression postoperative outcome regarding OS and CS, due to the low number of positive samples (Appendix A). 

### 2.2. BHLHE41 Expression Is Associated with Early-Stage Cancer

In addition, among the clinicopathologic findings in LUAD cases, the expression of BHLHE41 protein was also associated with AIS histology (*p* < 0.001), lack of pleural invasion (*p* = 0.026), T factor (T1) (*p* = 0.013), and stage (IA) (*p* = 0.013) (Table 2). 

Gender, tumor size, pleural invasion, T factor, N factor, and stage were significant predictors of better OS in the univariate survival analysis in LUAD, however BHLHE41 expression was not that in the univariate and multivariate analysis, possibly because the positive cases of the samples were limited (Table 3). In the multivariate analysis, the female sex constituted a factor correlating with better survival (*p* =0.002), as reported previously [19,20].

In contrast, BHLHE41 expression in non-LUAD cases did not significantly correlate with any clinicopathologic factors and was not a prognostic factor (Appendix A and Table 3). These results appear to be partly consistent with the in silico results. However, these differences between LUAD and non-LUAD samples might be explained by the smaller number of positive samples in the non-LUAD group, possibly because early LUSC tumors within the bronchus are only rarely acquired as surgical resection samples, and the 45 samples did not contain early cancer, partly due to difficulties in identifying early-stage LUSC and the highly curative effect of photodynamic therapies [21].

To estimate whether BHLHE41 is expressed in early-stage LUSC tissues, we examined 18 early LUSC tissues using IHC staining (Figure 2h,i). Among 18 tumors, 7 were positively stained, yielding a positive ratio (38.9%) much higher than that in the non-LUAD group (4.4%) (Table 4 and Appendix A). In early LUSC tissues, BHLHE41 protein expression was significantly inversely correlated with the depth of invasion (DI) on the basis of bronchial tissue structures (*p* = 0.0048), as shown using the Mann–Whitney U test (Table 4) [22]. These data suggest that BHLHE41 expression decreases in invasive cancer cells not only in LUAD but also in LUSC.

### 2.3. BHLHE41 Expression Suppresses Cell Proliferation and Induces Autophagy 

To assess BHLHE41 function in LUAD, we performed a survival assay, revealing that DOX could suppress the proliferation of DOX-inducible BHLHE41-expressing A549 and H2030 cells but not control cells in 3 days (Figure 4a,b). In the xenograft model, the A549/BHLHE41#1 DOX intake group exhibited smaller tumor sizes and lower weight than the non-DOX intake group (Figure 4c,d).

In the DOX-inducible BHLHE41-expressing A549 and H2030 cells, BHLHE41 expression did not upregulate cleaved caspase 3, cleaved polyADP ribose polymerase (PARP), or phosphorylated γH2AX as shown using immunoblotting (Appendix A); moreover, the caspase 3 inhibitor Z-VAD-FMK could not rescue the growth suppression of these cells (Appendix A). These data suggest that BHLHE41 expression did not induce DNA damage or apoptosis. In contrast, we observed LC3-II accumulation in these cells 2 days following DOX treatment (Figure 5a–c), and the lysosomal inhibitors chloroquine (CQ) (Figure 5a) or bafilomycin A1 (Figure 5b) enhanced LC3-II accumulation. These results indicate that BHLHE41 expression induced autophagosome production, fusion with lysosomes, and degradation. Moreover, using the tandem fluorescence-tagged LC3 (ftLC3) plasmid, we could distinguish autophagosomes that were fused and not fused to lysosomes, as GFP is sensitive to acidic pH in lysosomes but mRFP is not. In merged images, red and yellow punctate spots indicate LC3 fused and not fused to lysosomes, respectively. In A549/BHLHE41#1 and #2 cells, the numbers of both red and yellow spots increased in the presence of DOX compared with those in its absence; however, they were comparable in A549/control#1 under either condition [23] (Figure 5d and Appendix A). Our results showed an increased number of autophagosomes which are degraded following fusion with lysosomes, influenced by BHLHE41 expression and supported the hypothesis that autophagy is enhanced upon BHLHE41 induction in lung cancer cells. Notably, CQ could rescue the suppression of proliferation of BHLHE41-expressing A549 and H2030 cells (Figure 4a,b).

## 3. Discussion

Lung cancer constitutes a serious global health problem. Several driver mutations of oncogenes encoding, for instance, EGFR, K-Ras, HER2, and BRAF, and fusion genes encoding ALK, RET, and ROS have been identified in LUAD, with changes in these genes serving as indicators for the use of molecular targeted drugs [4]. Recently, genomic, epigenetic, and transcriptomic changes in carcinoma in situ of LUSC have also helped understanding LUSC progression [24]. Nevertheless, although patients with pathological stage I NSCLC obtain considerable benefit from surgical resection, prognostic heterogeneities in stage I NSCLC have been reported, not only depending on sex and host immune and nutritional status factors [25,26], but also on histological classification with or without lepidic growth in LUAD [27,28,29,30] and angiolymphatic and pleural invasion [31]. In particular, patients with LUAD and AIS lung cancer have a favorable prognosis compared to those with advanced disease [30], while patients with invasive adenocarcinoma and lepidic growth have a better prognosis than those without lepidic growth [27]. It is therefore crucial to identify molecules to stratify patients with early lung cancer according to prognoses to achieve their curative treatment in addition to understanding the mechanism underlaying early progression. Previously, it was reported that Stratifin, USP9, and CXCL10 are associated with the progression from AIS to early invasive adenocarcinoma [32,33,34], whereas an increase in Notch2 and Six1 and disappearance of RhoB are related to progression from lepidic growth to early invasive cancer in LUAD [35,36]. Jacobsen et al. have further reported that C4.4A is expressed in precancerous lesions in LUAD and LUSC and is related to poor prognosis of LUAD [37,38]. In the present study, BHLHE41 was only expressed in low-invasive lesions in both LUAD and LUSC cells, although we did not find any association between BHLHE41 and the previously reported molecules described above. 

Autophagy is a double-edge sword in cancer development and treatment because the acceleration of this process can not only suppress cancer development but also promote survival under conditions of strict microenvironment and cancer chemotherapy [39,40]. However, many researchers have recently considered autophagy as a good therapeutic target for cancer [40,41]. It has been reported that BHLHE41 suppresses expression of FAS, BAX, C-MYC, and inhibit apoptosis of MCF7 cells by TNF treatment [13]. In addition to BAX, induction of BHLHE41 reduced BCL2, which has been reported to suppress autophagy by interacting with BECLIN [42]. Suppression of BCL2 function by BHLHE41 expression might be one of the mechanisms by which BHLHE41 induces autophagy cell death. The precise mechanisms of autophagy induction by BHLHE41 expression needs to be elucidated. The balance of BHLHE41 and BHLHE40 expression and their molecular interaction might influence their roles in cancer development. Our results suggested that the selective autophagy induced by BHLHE41 in lung cancer cells could inhibit lung cancer progression. In contrast, BHLHE41 was reported to be related to tumor progression in MLL-AF6-positive AML and renal clear-cell carcinoma [16,17,43]. In MLL-AF6 positive AML cells, BHLHE41 has been reported to interact with MLL-AF6 and activate several leukemogenic genes, such as FOXD4L1, CDK6, and ZNF521 [17]. Our results show that BHLHE41 has not only a suppressive but also an activation function as a transcription factor that depends on the background gene expression. Taken together, these data suggest that BHLHE41 plays an important role in inducing autophagic cell death during early lung cancer cells and likely suppresses lung cancer progression. BHLHE41 can significantly influence the development and can be used a biomarker of invasiveness of NSCLC. It remains to be elucidated as to how the autophagic cell death cascade is activated in cancer cells but not in normal cells by BHLHE41 expression.

## 4. Materials and Methods

### 4.1. In Silico Analysis of BHLHE41 Expression in Cancer

Comparison of BHLHE41 RNA expression between normal tissue and LUAD or LUSC using TCGA data was performed using UALCAN (http://ualcan.path.uab.edu/, accessed on 30 May 2021) [44]. To determine the relationship between BHLHE41 expression and prognosis, RNA-Seq bam files aligned to the human genome (hg38) and survival information of patients with LUAD in the TCGA project were downloaded from the Genomic Data Commons portal site (https://portal.gdc.cancer.gov, accessed on 17 July 2018). These files were analyzed using a series of cufflinks software programs, cuffquant and cuffnorm, to quantify and normalize the expression of the BHLHE41 gene. Patients were divided into two groups based on the best cut-off point: high and low BHLHE41 expression; then, a survival curve was constructed via the Kaplan–Meier method using survfit (R package). *p*-values were calculated from the log-rank test using survdiff (R package). 

### 4.2. Reagents and Antibodies

The following reagents were purchased from the indicated manufacturers: CQ and bafilomycin A1 (FUJIFILM Wako Pure Chemical Industries, Osaka, Japan), Z-VAD-FMK (Peptide Institute, Osaka, Japan), and 3-(4,5-dimethylthiazol-2-yl)-2,5-diphenyl tetrazolium bromide (MTT) and doxycycline (DOX) (Sigma-Aldrich, St. Louis, MO, USA). Antibodies against BHLHE41 (H-72, sc-32853 and E-4, sc-373763), PARP1 (sc-8007), β-actin (BMR00270; Bio Matrix Research, Nagareyama, Japan), caspase 3 (9622; Cell Signaling Technology, Danvers, MA, USA), phosphor-γH2AX (phospho S139) (ab11174; Abcam, Cambridge, UK), and LC-3 (PM036; MBL, Nagoya, Japan) were purchased. Horseradish peroxidase-conjugated anti-rabbit or anti-mouse IgG (7074S, 7076S, Cell Signaling Technology) was used as the appropriate secondary antibody.

### 4.3. Patients and Tumor Samples 

This study included 177 consecutive patients with NSCLC who underwent surgical treatment, i.e., lobectomy and segmentectomy, in the Department of General Thoracic Surgery of Kagoshima University Hospital between January 2009 and December 2010. Clinical samples were obtained from tumors that were surgically removed and pathologically confirmed as NSCLC. The patients consisted of 96 males and 81 females aged between 38 to 86 years (Table 1). No patient had received preoperative radiotherapy or chemotherapy. The pathological features of NSCLC were defined according to the Tumor–Node–Metastasis (TNM) classification 7th edition [45] and histological evaluation following UICC TNM classification 8th edition [2]. Written informed consent was obtained from each patient. A total of 18 early LUSC samples were resected at Tohoku University. These studies were approved by the institutional review board of Kagoshima and Tohoku University and performed in accordance with the Helsinki Declaration.

### 4.4. IHC and Assessments

The surgical samples were fixed in 10% phosphate-buffered formalin, embedded in paraffin, cut into 3 μm slices, and mounted on glass slides. Specimens were then incubated in 0.01 M sodium citrate (pH 6.0) for 10 min, 3% hydrogen peroxide for 30 min, and 1% bovine serum albumin for 30 min. Sections were then incubated in a humidified chamber at 4 °C overnight with BHLHE41 rabbit polyclonal antibody (1:150 dilution), stained with diaminobenzidine tetrahydrochloride (DAKO, Glostrup, Denmark) using the avidin–biotin complex and immunoperoxidase method (Vectastatin ABC Kit, Vector Laboratories, Burlingame, CA, USA), and counterstained with hematoxylin. BHLHE41 expression was determined by counting the number of cancer cells in which the nucleus was stained with the anti-BHLHE41 antibody. Evaluation of IHC was independently carried out by two board-certified pathologists (TH and AT), and the inconsistent cases were reevaluated under agreement. Ten fields within the tumors were selected and staining in 1000 cancer cell nuclei (100 cells per field) was observed at ×200 magnification using microscopy. The samples were graded as BHLHE41-positive if more than 25% of cancer cells were stained and as BHLHE41-negative if <25% of cancer cells were stained. 

### 4.5. Correlation between the DI and BHLHE41 Staining of Early LUSC Cases

The relationship between BHLHE41 expression and the DI was assessed using the Mann–Whitney U test. DI was scored on the basis of bronchial tissue structures as previously described: DI, 0, carcinoma in situ; 1, suspicious invasion; 2, intramucosal invasion; 3, extramuscular invasion; 4, extracartilaginous invasion; 5, extra bronchial invasion [20].

### 4.6. Cell Lines and Culture

A549 and H2030 adenocarcinoma cells were obtained from the American Type Culture Collection (Manassas, VA, USA). These cells were cultured in DMEM and RPMI-1640 from Nissui Pharmaceutical (Tokyo, Japan), respectively, with 10% fetal bovine serum (Biosera, Kansas City, MO, USA) at 37 °C in a humidified atmosphere of 5% CO_2_. 

### 4.7. Vectors

S-IV-TRE-RfA-UbC-Puro was kindly provided by Dr. Johmura (Univ. Tokyo. Inst. Med Research) [46]; pCMV-VSV-RSV-Rev and pCAG-HIVgp were purchased from the RIKEN BioResource Center (Tsukuba, Japan). The ftLC3 plasmid encoding mRFP-GFP-LC3 was produced following a method previously reported [21].

### 4.8. Establishing BHLHE41-Expressing Cells

Human BHLHE41 cDNA was isolated as previously described [7], inserted into pENTR-1A, mixed with CS-IV-TRE-RfA-UbC-Puro vector, and reacted with Gateway LR clonase to generate the lentivirus plasmid. Lentiviruses expressing DOX-inducible BHLHE41 were generated by cotransfection of the lentivirus plasmid with pCMV-VSV-RSV-Rev and pCAG-HIVgp into HEK293T cells. A549 and H2030 cells were infected with the virus and selected using 10 μg/mL puromycin for 2 days. A549/BHLHE41#1, #2, and H2030/BHLHE41#1, #2, and A549/control#1 and H2030/control#1 cells were established via empty vector virus transfection. DOX was added to the medium for A549 and H2030 parental and derived cells at concentrations of 0.5 and 2 μg/mL, respectively, to induce BHLHE41 expression. 

### 4.9. Cell Proliferation Assay 

Equal numbers of cells (1 × 10^3^) were seeded into 96-well plates and incubated for 3 days. Cell viability was measured using the MTT colorimetric assay, as described previously [47].

### 4.10. Protein Extraction and Immunoblotting

Total cell lysates were isolated using RIPA buffer (25 mM Tris-HCl pH 7.5, 150 mM NaCl, 1% Nonidet P-40, 0.1% sodium dodecyl sulfate (SDS), 1% sodium deoxycholate) with a proteinase inhibitor cocktail (Nacalai Tesque, Kyoto, Japan). Protein concentrations were measured using a Protein Assay CBB Solution (5×) (Nacalai Tesque). Immunoblotting was performed, as previously described [48]. Cell lysates (20 μg protein) were separated using SDS-polyacrylamide gel and transferred onto polyvinylidene fluoride membranes (Merck, Darmstadt, Germany). The blotted membranes were incubated with the indicated primary antibodies overnight at 4 °C and the appropriate secondary antibody at 25 °C for 1 h, then each protein was detected using the clarity Western ECL substrate (Bio-Rad, Hercules, CA, USA). The levels were evaluated using Image J software (NIH, Bethesda, MD, USA).

### 4.11. Observation of Fluorescence Associated with ftLC3 Expression

mRFP- and GFP-tagged LC3 expression was observed, as previously reported [21]. In brief, 2 × 10^5^ cells were plated on 12-mm cover slips and cultured for 24 h, after which ftLC3 plasmid was transfected via lipofection. After 24 h, cells were transferred to fresh media with or without DOX. Media were removed, and cells were fixed using 4% paraformaldehyde in phosphate-buffered saline (PBS) for 15 min. After removing the fixation solution and washing with PBS, the cells were incubated in 50 mM NH_4_Cl in PBS for 10 min. After washing with PBS, the cover slip was placed on a glass slide. The cells were examined under an LSM780 fluorescence laser scanning confocal microscope (Zeiss, Oberkochen, Germany) at 60× magnification. 

### 4.12. Xenograft Model Experiments and Animal Husbandry

Female BALB/c nude mice were obtained from CLEA Japan (Tokyo, Japan). A549 and A549 BHLHE41#1 cells (1 × 10^6^) were transplanted subcutaneously into nude mice. Following tumor growth to 100 mm3, the mice transplanted with each cell line were separated randomly into two groups and fed ad libitum with 5% sucrose water with or without DOX (2 mg/mL). The tumor volumes were calculated using the length × width × height of tumors and the size was measured every week. On day 35 after separating the groups, the mice were sacrificed, and the isolated tumors were weighed. All mice were housed under a 12-h dark–light cycle (light from 07:00 to 19:00) at 22 ± 1 °C with ad libitum food and water. The animal experiments were conducted according to the protocol approved by the Institutional Animal Care and Use Committee of Kagoshima University. All experiments were conducted in compliance with the ARRIVE guidelines.

### 4.13. Statistical Analysis

Survival curves from IHC analyses were plotted using the Kaplan–Meier method and compared using the log-rank test via SPSS (IBM, Armonk, NY, USA). Univariate and multivariate Cox regression analyses were performed by SPSS. Statistical analyses for all experiments with two groups were performed using the Student’s *t*-test and those including more than three groups were evaluated using one-way analysis of variance with ad hoc Bonferroni tests. Data are presented as the means ± SD. Differences were considered significant at *p* < 0.05.

## Figures and Tables

**Figure 1 ijms-22-11509-f001:**
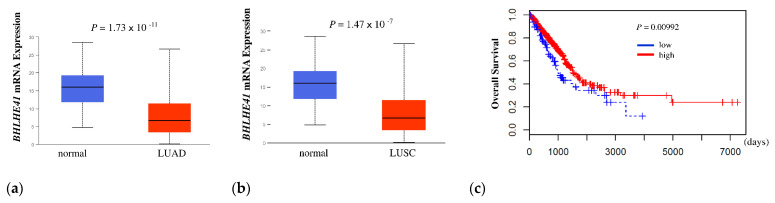
In silico analysis of *BHLHE41* expression and Kaplan Meier plot data. (**a**) In silico analysis of expression of *BHLHE41* in LUAD (*n* = 515) and normal lung tissue (*n* = 59); and (**b**) LUSC (*n* = 503) and normal lung tissue (*n* = 52) from TCGA database. (**c**) The Kaplan–Meier curve was plotted from TCGA data of patients with LUAD, as described in the Material and Methods section. Red: Higher expression group (402 patients), Blue: lower expression group (98 patients). X-axis indicates days and Y-axis indicates overall survival.

**Figure 2 ijms-22-11509-f002:**
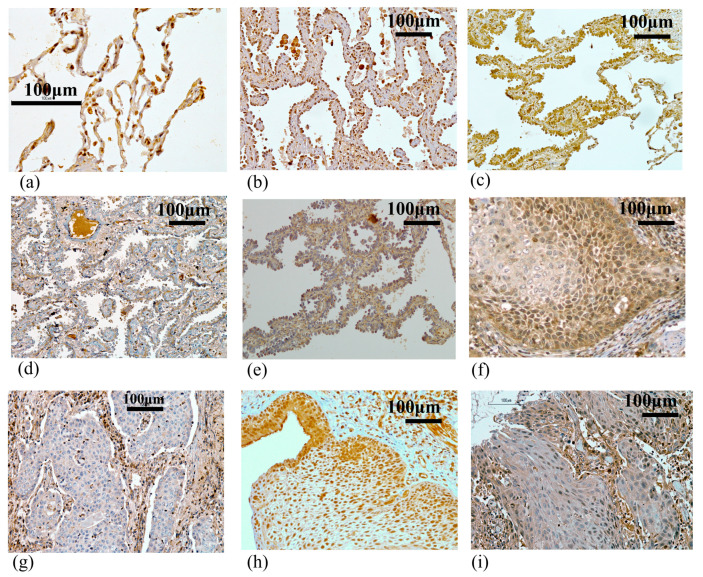
Immunochemical staining using the anti-BHLHE41 antibody. (**a**) Normal alveolar tissue (×400). (**b**) Positively stained case of adenocarcinoma in situ: p stage 0: TisN0M0. (**c**) Positively stained case of the lepidic growth part, and (**d**) negatively stained case of the invasive growth part in papillary adenocarcinoma (lepidic growth 40%): p stage IIIA: T4N1M0. (**e**) Negatively stained case of the lepidic growth part in papillary adenocarcinoma (lepidic growth 5%): p stage IIIA: T3N2M0. (**f**) Positively stained case of squamous cell carcinoma: p stage IA: T1aN0M0. (**g**) Negatively stained case of squamous cell carcinoma: p stage IA: T1aN0M0. (**h**) Positively stained case of early squamous cell carcinoma: DI:2. (**i**) Negatively stained case of early squamous cell carcinoma: DI:5. All pictures were taken at a magnification of 200× (except for A:×400); scale bar, 100 μm.

**Figure 3 ijms-22-11509-f003:**
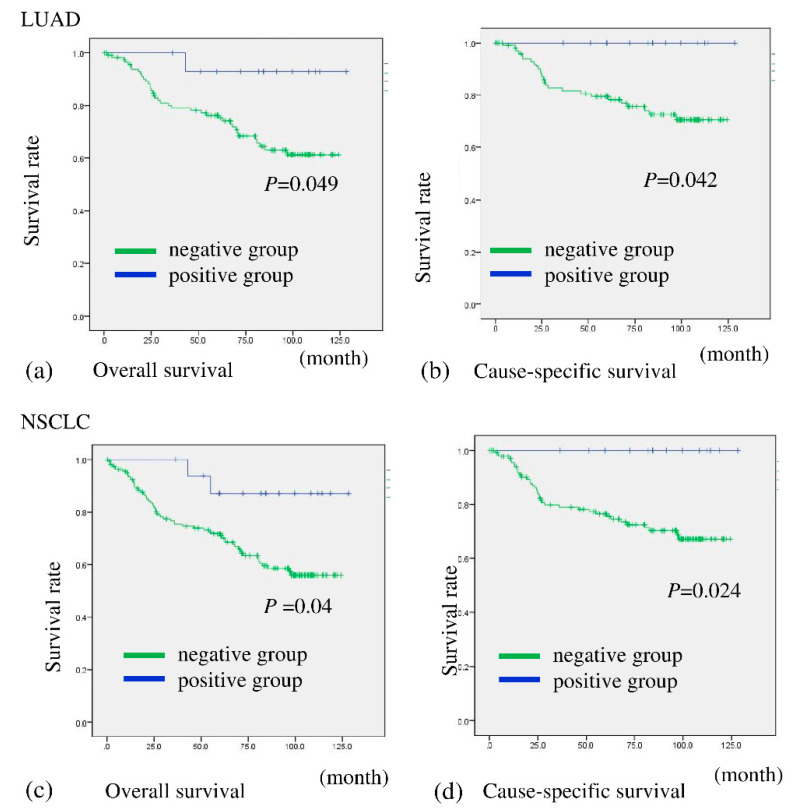
Overall survival and cause-specific survival of patients according to their expression of BHLHE41 in NSCLC and LUAD. Patients with BHLHE41-positive LUAD had significantly more favorable prognoses than those with BHLHE41-negative LUAD for OS (*p* = 0.049) (**a**) and CS (*p* = 0.042) (**b**). Patients with BHLHE41-positive NSCLC had significantly more favorable prognoses than those of BHLHE41-negative patients for OS (*p* = 0.04) (**c**) and CS (*p* = 0.024) (**d**).

**Figure 4 ijms-22-11509-f004:**
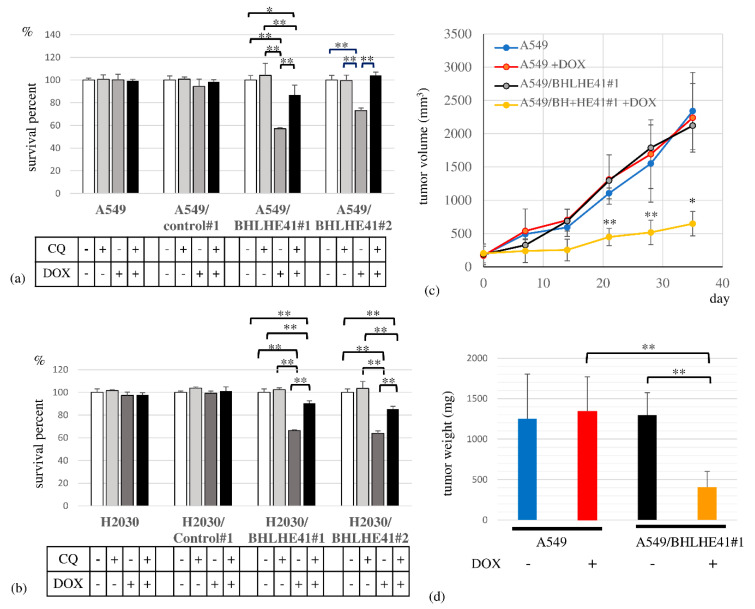
Cell survival assay and xenograft model experiment. Cell survival assay in the presence and absence of CQ (10 μM) or DOX for A549- and A549-derived cells (0.5 μg/mL), and for H2030- and H2030-derived cells (2 μg/mL) for 3 days. (**a**) A549 cells, A549/control#1, A549/BHLHE41#1 and A549/BHLHE41#2 cells; (**b**) H2030; H2030/control#1; H2030/BHLHE41#1 and H2030/BHLHE41#2 cells. (**c**) In the xenograft model, the growth of tumors of A549 and A549/BHLHE41#1 cells with and without DOX (2 mg/mL)-containing water. Statistical significance of the difference between A549/BHLHE41#1 tumors with and without DOX water-based feeding is indicated as * *p* < 0.05 at the 35th day and ** *p* < 0.01 at the 21st and 28th day. (**d**) Weight of isolated tumors of A549 and A549/BHLHE41#1 cells with and without feeding of DOX-containing water.

**Figure 5 ijms-22-11509-f005:**
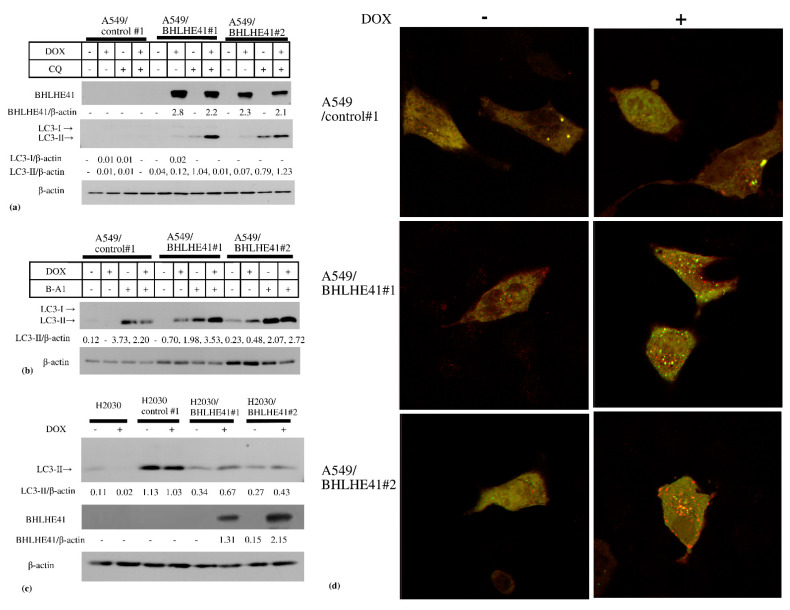
Immunoblotting of autophagy-related proteins. Immunoblotting of autophagy-related proteins. (**a**) In A549 and derived cells, DOX induced the accumulation of LC3-II and CQ enhanced this accumulation. (**b**) In A549 and derived cells, DOX induced the accumulation of LC3-II and 5 nM bafilomycin A1(B-A1) treatment increased LC3-II accumulation. (**c**) In H2030 and derived cells DOX could increase LC3-II accumulation. (**d**) Merged photo of GFP and mRFP photos of fluorescence-tagged LC3 expression of A549/control#1, A549/ BHLHE41#1, and A549/BHLHE41 #2 transiently transfected with the ftLC3 plasmid and observed via confocal laser microscopy, after incubation with medium with or without DOX (0.5 μg/mL) for 48 h. The red spots indicate autophagosomes that is not fused with lysosomes and the yellow spots are autophagosomes fused with lysosomes.

**Table 1 ijms-22-11509-t001:** Clinicopathological factors in the lung cancer patients.

		Number	%
Gender	Male	96	54.2
	Female	81	45.8
Age	Median	70	
	Range	38–86	
Operation	Pneumonectomy	2	1.1
	Lobectomy	154	87.0
	Segmentectomy	21	11.9
p stage	IA	83	46.9
	IB	38	21.5
	IIA	22	12.4
	IIB	11	6.2
	IIIA	23	13.0
Histology	Adenocarcinoma	132	74.6
	Squamous cell carcinoma	43	24.3
	Others	2	1.1

**Table 2 ijms-22-11509-t002:** Correlation between BHLHE41 expression and clinicopathologic factors in LUAD patients.

Clinicopathologic Factors		Expression of BHLHE41				
		Positive	%	Negative	%	*p* Value
		*n* = 15	(11.4)	*n* = 117	(88.6)	
Age≥	<70 years	8	(13.6)	51	(86.4)	0.475
	≥70 years	7	(9.6)	66	(90.4)	
Gender	female	10	(13.0)	67	(87.0)	0.487
	male	5	(9.1)	50	(90.9)	
Tumor size	≤30 mm	13	(14.6)	76	(85.4)	0.076
	>30 mm	2	(4.7)	41	(95.3)	
Pleural invasion	No	15	(14.3)	90	(85.7)	0.026 *
	Yes	0	(0.0)	27	(100.0)	
Pulmonary metastasis	No	15	(12.0)	110	(88.0)	0.421
	Yes	0	(0.0)	7	(100.0)	
T factor	T1	14	(16.5)	71	(83.5)	0.013 *
	≥T2	1	(2.1)	46	(97.9)	
N factor	N0	13	(12.0)	95	(88.0)	0.46
	N1/2	2	(8.3)	22	(91.7)	
Stage	IA	13	(17.3)	62	(82.7)	0.013 *
	≥IB	2	(3.5)	55	(96.5)	
Histology	AIS	7	(46.7)	8	(53.3)	<0.001 *
	Invasive	8	(6.8)	109	(93.2)	

Histological classification AIS (adenocarcinoma in situ) was followed to TNM classification 8th edition [2]. * indicates significant correlation (*p* < 0.05).

**Table 3 ijms-22-11509-t003:** Univariate and multivariate COX regression analysis in the patients with LUAD for overall survival.

Clinicopathological Factors					Univariate		Multivariate	
		*n*	Dead	Alive	HR (95%CI)	*p* Value	HR (95%CI)	*p* Value
Age	≥70 years	73	25	48	1.70	0.11	1.53	0.21
	<70 years	59	14	45	(0.88–3.28)		(0.79–2.96)	
Gender Male	male	55	25	30	3.33	<0.001 *	2.91	0.002 *
	female	77	14	63	(1.72–6.43)		(1.49–5.66)	
Tumor size	>30 mm	43	19	24	2.11	0.02 *		
	≤30 mm	89	20	69	(1.13–3.96)			
Pleural invasion	Yes	27	13	14	2.56	0.006 *		
	No	105	26	79	(1.32–4.99)			
Pulmonary metastasis	Yes	7	3	4	1.69	0.39		
	No	125	36	89	(0.52–5.48)			
T factor	≥T2	47	20	27	2.06	0.024 *		
	T1	85	19	66	(1.10–3.85)			
N factor	N1/2	24	15	9	3.95	<0.001 *		
	N0	108	24	84	(2.07–7.56)			
Pathological stage	≥IB	57	27	30	3.60	<0.001 *	2.69	0.005 *
	IA	75	12	63	(1.82–7.11)		(1.34–5.38)	
Histology	invasive	117	39	78	24.46	0.14		
	AIS	15	0	15	(0.35–1709)			
BHLHE41	negative	117	38	79	5.82	0.082	3.67	0.20
	positive	15	1	14	(0.80–42.39)		(0.49–27.35)	

In multivariate analysis, number of factors was determined by one tenth of dead cases and factors were select in considering multicollinearity. Histological classification AIS (adenocarcinoma in situ) was followed to TNM classification 8th edition [2]. * indicates significant correlation (*p* < 0.05).

**Table 4 ijms-22-11509-t004:** Correlation of BHLHE41 expression and DI of early LUSQ.

DI: Depth of Invasion		Expression of BHLHE41	
	*n*	Positive	Negative
DI 1	0	0	0
DI 2	2	2	0
DI 3	6	4	2
DI 4	7	1	6
DI 5	3	0	3
Total	18	7	11

Depth of invasion is indicated as DI. 0: carcinoma in site, 1: suspicious invasion, 2: intramucosal invasion, 3: extramuscular invasion, 4: extracartilaginous invasion, and 5: extrabronchial invasion, respectively. DI and staining were significantly correlated by Mann–Whitney U test (*p* = 0.0048).

## Data Availability

The datasets generated or analyzed during the current study are available from the corresponding author on reasonable request.

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
