# Peer review of "BHLHE41/DEC2 Expression Induces Autophagic Cell Death in Lung Cancer Cells and Is Associated with Favorable Prognosis for Patients with Lung Adenocarcinoma"

_ijms, 2021, doi:10.3390/ijms222111509_

Round 1
Reviewer 1 Report
- The rational of the developing mechanism underlying the actions of BHLHE41 involved in autophagic cell death should be included in the introduction and any references related to this process can be added.
- Interesting results regarding BHLHE41 did not alter the activation of caspase-3, increase the cleaved PARP, instead inducing the LC-3 accumulation. The BHLHE41 regulated proteins which can interact the proteins, such as LC-3 or any other proteins in autophage pathways should be considered to include in the results.
- Figure 5 B, The actin loading controls are really varied, suggest to repeat this set experiments to get the equal loading control level.
- Figure 5 D need to increasing cell numbers and including the scare bar in the figure, the second cell line, such as H2030 should be considered to be including in the results.
- The BHLHE41 sensitized autophage pathway is interesting findings, however, the detail molecular mechanism can be investigated more such as transcriptom, proteomics and protein interaction net work involved in autophage pathway will be enhancing the value and attractive to general readers.
Author Response
Reviewer1
Comments and Suggestions for Authors
- The rational of the developing mechanism underlying the actions of BHLHE41 involved in autophagic cell death should be included in the introduction and any references related to this process can be added.
Response:
I appreciate the reviewers comments. BHLHE41 has not been reported to have relation with autophagy and so far we could not have clarified the exact mechanisms of autophagy induced by BHLHE41 expression. However BHLHE41 can inhibit BAX and BCL2 expression might be relate to the function. So I judged it is proper to add our hypothesis in the discussion with references as follow “It has been reported that BHLHE41 suppresses expression of FAS, BAX, c-Myc and inhibit apoptosis of MCF7 cells by TNF treatment [13]. In addition to BAX, induction of BHLHE41 reduced BCL2, which has been reported to suppress autophagy by interacting with BECLIN [42]. Suppression of BCL2 function by BHLHE41 expression might be one of the mechanisms by which BHLHE41 induces autophagy cell death. The precise mechanisms of autophagy induction by BHLHE41 expression needs to be elucidated.“
- Interesting results regarding BHLHE41 did not alter the activation of caspase-3, increase the cleaved PARP, instead inducing the LC-3 accumulation. The BHLHE41 regulated proteins which can interact the proteins, such as LC-3 or any other proteins in autophagy pathways should be considered to include in the results.
Response:
I appreciate the reviewers comments. We had some results of the other proteins but still the research about the mechanisms of autophagy induced by BHLHE41 are in progression. I hope all these results should show in the next report and here I’d rather focus on the result that BHLHE41 expression is a tumor suppressor in early lung cancer.
- Figure 5 B, The actin loading controls are really varied, suggest to repeat this set experiments to get the equal loading control level.
Response:
I appreciate the reviewers comments. I understand the point. However, beta actin amount might not be same in different cells in some time, after standardized protein amount of cell lysate. We can still clearly indicate the difference expression of LC3 and these differences cannot change the conclusions. In addition, unfortunately we are supposed to response quickly to the reviewer and don’t enough time to repeat the experiments.
- Figure 5 D need to increasing cell numbers and including the scare bar in the figure, the second cell line, such as H2030 should be considered to be including in the results.
Response:
I appreciate the reviewers comments. I understand the point. I could get the consistent result with LC3 expression in Fig.5C in Fig.5D and I am sorry but we could not get enough number fluorescence expressing cells in some cell lines so I hope the suggested experiments will be done as further experiments.
5.The BHLHE41 sensitized autophagy pathway is interesting findings, however, the detail molecular mechanism can be investigated more such as transcriptome, proteomics and protein interaction network involved in autophagy pathway will be enhancing the value and attractive to general readers.
Response:
I appreciate the reviewers comments. I agree with the reviewer’s comments. To make clear the mechanisms of autophagy induction by BHLHE41, We are continuing experiments and need a lot of more experiments to make clear the point. In this manuscript we’d like to focus on the disappearance of BHLHE41 in lung cancer is an early cancer indicator.

Reviewer 2 Report
I do not have any suggestion or comment
Author Response
We appreciate the reviewer's estimation about out manuscript.
Reviewer 3 Report
In the present study by Nagata et al. the authors identify a loss of BHLHE41 as a novel marker of advanced lung cancer. The authors make great use of multiple model systems. Overall, this study provides a unique idea on the role of BHLHE41 in cancer cells. Below find a few comments that may strengthen the study.
Major:
- The introduction lacks some detail
- For Figure 1c, how did the authors determine low vs high expression, what were the parameters used to decide the cutoff of expression?
- It is not immediately clear what the authors what samples are included in the samples labelled as “non-adenocarcinoma.”
- As with figure 1c, for Figure 3, how did the authors determine positive vs negative staining? What was the threshold?
- Figure S2 can benefit from a larger n, it is difficult to conclude that “BHLHE41 expression was not significantly associated with postoperative outcome regarding OS and CS”.
- It would be helpful to know what was used as the positive controls in western blots, are these additional samples used or peptides?
- In order to make comparisons such as “Our results showed an increased number of autophagosomes” please provide quantification of the IF.
- The discussion should be expanded to have include a more comprehensive interpretation of the results.
Minor
Figure 1, use distinguishing colors for LUAD and LUSC
Figure 2e is out of order in the figure legend
Figure 4a and b need y-axis labels
In text citation of Western blots (line 180) looks to be mislabeled
Author Response
Reviewer 3
Major:
- The introduction lacks some detail.
Response:
I appreciate the reviewer’s comments. I agree with the comment and add more detail information about BHLHE41 in the introduction.
“BHLHE41 functions as a negative regulator of circadian rhythm suppression of Clock/Bmal1, Per expression, similar to BHLHE40 [8] and regulates sleep duration. DEC2(P385R) genotype is associated with a human short sleep phenotype [9]. In cell differentiation, BHLHE41 can interact with C⁄EBPα and β, inhibit their transcriptional activity and suppress adipogenesis in adipocyte precursor cells [10]. BHLHE41 has been also reported to express selectively in Th2 cells among Th lineage and indispensable to Th2 differentiation in mice [11]. The regulation of expression of multiple genes by BHLHE40 and BHLHE41 also has been reported to affect cancer development in apoptosis, epithelial mesenchymal transition, and hypoxic reaction [12]. BHLHE41 expression suppresses apoptosis induced by TNFα in breast cancer cells[13], twst1 expression and migration in human endometrial cancer cells [14], is also associated with better prognosis of patients with breast cancer with HIF1 protein suppression [15]. It has also been reported to have oncogenic roles in renal clear-cell cancer and MLL-AF6-positive acute myelogenous leukemia (AML) with other transactivating proteins [16, 17]. From these previous studies BHLHE41 likely can work as both a tumor promoter and sup-pressor dependently on cellular context. BHLHE41 also has been reported to act as a tumor suppressor gene in the lung [18] however, the roles of BHLHE41 in lung cancer development remain unclear. In the present study, we therefore examine the correlation between BHLHE41 expression and the prognosis of patients with NSCLC and explored the potential roles of BHLHE41 as a tumor suppressor during lung cancer development and progression.”
- For Figure 1c, how did the authors determine low vs high expression, what were the parameters used to decide the cutoff of expression?
Response:
I appreciate the reviewer’s comments. We separated into high and low expression group on the basis of the best cut-off point as described “Material and Method”. More practically at first we evaluated in median, lower 80% and higher 20%, and lower 20% (100cases) and higher 80% (400cases) group, we got significant P value (P=0.0111) in lower 20% (100 patients) and higher 80% (400 patients). To find less P values we check around the data to lower 98 patients and higher 402 patients as shown in Fig. 1C.
- It is not immediately clear what the authors what samples are included in the samples labelled as “non-adenocarcinoma.”
Response:
I appreciate the reviewer’s comments. Non-adenocarcinoma include as indicated in Table 1. Including 43 cases of SqCC and 2 cases of pleomorphic carcinoma. To make clear that I add the sentence “including 43 SqCC and 2 pleomorphic carcinoma” in L 112.
- As with figure 1c, for Figure 3, how did the authors determine positive vs negative staining? What was the threshold?
Response:
I appreciate the reviewer’s comments. We separate the samples into positive and negative as described in material and methods like following sentences. “Ten fields within the tumors were selected and staining in 1,000 cancer cell nuclei (100 cells per field) was observed at ×200 magnification using microscopy. The samples were graded as BHLHE41-positive if more than 25% of cancer cells were stained and as BHLHE41-negative if < 25% of cancer cells were stained.” and “Evaluation of IHC was independently carried out by two board-certified pathologists (TH and AT), and the inconsistent cases were reevaluated under agreement.”
- Figure S2 can benefit from a larger n, it is difficult to conclude that “BHLHE41 expression was not significantly associated with postoperative outcome regarding OS and CS”.
Response:
I appreciate the comment of the reviewer. I agree with the idea of review. We meant the analysis didn’t indicate significant correlation. I changed the sentence like “it was difficult to evaluate association between BHLHE41 expression postoperative outcome regarding OS and CS, due to the low number of positive samples (Figure S2).” In line 123.
- It would be helpful to know what was used as the positive controls in western blots, are these additional samples used or peptides?
Response:
I appreciate the reviewer’s comments. I understand the reviewer’s idea. We show positive control samples in caspase 3, because we couldn’t detect the bands. Anit-BHLHE41 was evaluated with BHLHE41consistent expressing A549 cells and I attached western data. We used only well-established antibodies with other journal data.
- In order to make comparisons such as “Our results showed an increased number of autophagosomes” please provide quantification of the IF.
Response:
I appreciate the comment of the reviewer. We tried but couldn’t get enough number of staining cells in some cell lines. I hope I’d like to do that as a further experiment.
- The discussion should be expanded to have include a more comprehensive interpretation of the results.
Response:
I add some hypothesis about autophagy induction by BHLHE41.
Minor
Figure 1, use distinguishing colors for LUAD and LUSC
Response:
I appreciate the comment of the reviewer. However I am sorry we can’t change the color because this data fixed by application on web.
Figure 2e is out of order in the figure legend
I appreciate the comment of the reviewer. I replace the order of photo and collected the order of 2d and 2e. in Fig and legend.
Figure 4a and b need y-axis labels
I appreciate the comment of the reviewer. I added the label as “survival percent”.
In text citation of Western blots (line 180) looks to be mislabeled
I appreciate the comment of the reviewer. I added Figure S3a, b to after immunoblotting.
